# Effects of Sn Promoter on the Ordered Mesoporous Co₃O₄-Al₂O₃ Mixed Metal Oxide for Fischer–Tropsch Synthesis Reaction

**Dongming Shen †, Sang Beom Han †, Xu Wang, Mansoor Ali and Jong Wook Bae ***

School of Chemical Engineering, Sungkyunkwan University (SKKU), 2066 Seobu-ro, Jangan-gu, Suwon 16419, Republic of Korea

* Correspondence: finejw@skku.edu; Tel.: +82-31-290-7347; Fax: +82-31-290-7272

† These authors contributed equally to this work.

**Abstract:** The highly ordered mesoporous Co₃O₄-Al₂O₃ bimetal oxide, prepared by a nano-casting method, was modified with Sn promoter (denoted as Sn/m-CoAlOx) to enhance selectivity to liquid-hydrocarbons as well as to suppress $CO_2$ formation formed by a water gas-shift (WGS) reaction activity during CO hydrogenation to hydrocarbons (Fischer–Tropsch Synthesis (FTS) reaction). Based on the surface properties of the Sn/m-CoAlOx in the range of 0.25–0.65 wt%Sn, the Sn promoter generally decreased CO conversion and increased $C_{5+}$ selectivity through its non-selective blockages of the active metallic cobalt sites, which were responsible for more difficult reducibility of cobalt nanoparticles with an increase of Sn content as well. In addition to those contributions of Sn promoter, the decreased $CO_2$ and $CH_4$ selectivity was clearly observed on the optimal Sn(2)/m-CoAlOx with only small decrease of CO conversion with 79.1% from 81.5% for the reference m-CoAlOx. Those phenomena were mainly attributed to the suppressed WGS reaction activity as well as the decreased hydrogenation activity to form $CH_4$ due to the suppressed $H_2$ adsorption capacity on the less reduced surface Co sites on the Sn(2)/m-CoAlOx.

**Keywords:** Fischer–Tropsch Synthesis (FTS) reaction; mesoporous Co₃O₄-Al₂O₃ bimetal oxide; Sn promoter; product distribution; water-gas-shift (WGS) reaction

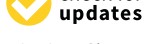

## 1. Introduction

Due to the depletion of the fossil fuel resources and environmental concerns, Fischer-Tropsch synthesis (FTS) reaction has been considered as one of the promising technologies to produce clean liquid fuels and value-added chemicals via syngas conversion derived from shale gas, natural gas, coal and biomass [1,2]. Among the active heterogeneous FTS catalysts, various Co-based catalysts have been industrially applied due to its relatively higher CO hydrogenation activity and higher selectivity to long-chain hydrocarbons with a lower competitive water gas shift (WGS) reaction activity [1,3]. The catalytic activity and product distribution were largely affected by particle sizes of the cobalt nanoparticles as well as types of supports and promoters, where the structure-insensitive natures of Co nanoparticles above 8 nm in size [4–6] suggested that the numbers of exposed metallic Co sites can be well correlated with the catalytic activity. In addition, on the various porous supporting materials such as Al₂O₃, SiO₂, TiO₂, zeolites and carbonaceous materials, the different metal-support interactions between those supports and active Co nanoparticles can largely change the structural and electronic properties of the surface Co nanoparticles. These results can be also responsible for the different degrees of aggregation of Co nanoparticles, reoxidation and heavy hydrocarbons depositions resulted in an easy deactivation [7–9]. Those deactivation phenomena can be effectively controlled by using some ordered mesoporous structures because of its larger specific surface areas and direct exposure of active metallic Co sites on the bimetal oxides framework surfaces. The

ordered mesoporous structures can be synthesized by the well-known nanocasting method, and the hard templates such as MCM-41, SBA-15 and KIT-6 can be used to prepare the highly ordered mesoporous materials as well [10–12]. Compared to the 2-dimensional wire-structures (p6mm) originated from the SBA-15 template, the 3-dimensional cubic porous structures (Ia3d) prepared from the KIT-6 template seem to be better to increase the number of active sites and smaller mass transfer limitations, which finally resulted in excellent catalytic performances such as $CO_2$ hydrogenation [13], oxidation of toluene and formaldehyde [14,15], methane combustion [16] and so on. However, based on our previous works [17–19], the mesoporous cobalt oxide itself was not stable during FTS reaction due to its facile structural collapses and easy phase transformations. Therefore, the irreducible metal oxides such as $Al_2O_3$ or $ZrO_2$ at optimal molar ratios of M:Co (M = Al or Zr) can be incorporated to form the thermally and chemically stable bimetal oxides with the help of partial formation of thermodynamically stable spinel-type phases such as Co $Al_2O_4$ and so on [20–22].

In addition, small amount of chemical or structural promoters can alter catalytic performances significantly, and various promoters having positive contributions of activity and stability such as noble metals, transition metals, alkali and alkali-earth metals and rare earth metals are generally incorporated on the cobalt-based FTS catalysts [23–25]. Among those promoters, Tin (Sn) has not been well-reported as a chemical promoter on the FTS catalysts as far as we know, even though Sn promoter has been largely applied for various reforming reactions [26,27], dehydrogenation of alkanes [28], $CO_2$ hydrogenation [29], dimethyl ether (DME) synthesis from syngas [30,31] and so on. The crucial roles of Sn promoter are to change the surface affinity as well as to modify the electronic states of active metals. For example, the Sn promoter can decrease the affinity of Ni nanoparticles for carbon depositions by largely enhancing the dissociation activity of $CH_4$ and $CO_2$ molecules with the suppressed adsorption strengths of surface carbons during dry reforming reaction [27]. Furthermore, Sn promoter can also change the electronic states, surface acidities and distributions of metallic Cu nanoparticles on Cu-ZnO-$Al_2O_3$ by suppressing the catalytic activity of water-gas shift (WGS) reaction as well as hydrocarbon formations during $CO_2$ hydrogenation to oxygenates [30,31].

In the present investigation, the highly ordered mesoporous m-CoAlOx catalyst was further modified with different amounts of Sn promoter for the possible applications of FTS reaction. According to the Sn contents on the Sn/m-CoAlOx, the different catalytic activity and product distribution were explained in terms of the strong metal-support interactions and electronic states of active cobalt nanoparticles with an insignificant decrease of CO conversion and enhanced $C_{5+}$ selectivity at an optimal Sn content less than 0.65 wt%.

## 2. Results and Discussion

### 2.1. Textural and Surface Properties of Sn/m-CoAlOx Catalysts

The contents of $SnO_2$ on the fresh Sn/m-CoAlOx were found to be close to their nominal values in the range of 0–1.82 wt% as confirmed by XRF results in Table 1. The highly ordered mesoporous structures of the Sn/m-CoAlOx were confirmed by a small angle X-ray scattering (SAXS) analysis as shown in Figure 1A, where the characteristic diffraction peaks appeared at ~0.9° suggest the well-defined ordered mesoporous structures on all the fresh Sn/m-CoAlOx. Those highly ordered mesoporous structures on the Sn/m-CoAlOx were also confirmed by $N_2$ adsorption–desorption analysis, which showed a typical type IV Isotherm with $H_2$ hysteresis as displayed in supplementary Figure S1. As summarized in Table 1, specific surface area on the Sn/m-CoAlOx was gradually decreased with an increase of Sn content from 109.7 $m^2/g$ on the unmodified m-CoAlOx to 92.2 $m^2/g$ on the Sn(18)/m-CoAlOx, which were attributed to the selective blockages of mesopore mouths by adding excess amount of Sn promotor. The average pore diameter on the fresh Sn/m-CoAlOx was found in the range of 3.8–4.1 nm (calculated from desorption branch of $N_2$-sorption isotherm) without any significant changes according to the Sn content, which were found to be a little smaller diameter than that of ~4.4 nm calculated from the

absorption branch due to the tensile strength and pore network effect [27]. In addition, pore volume was fiound to be also similar on all the fresh Sn/m-CoAlOx with 0.14 cm$^3$/g compared to the Sn(18)/m-CoAlOx with the smallest pore volume of 0.11 cm$^3$/g. As displayed in Figure 1B and supplementary Figure S1B, all the Sn/m-CoAlOx catalysts showed unimodal pore size distributions obtained from $N_2$-adsorption and desorption isotherm, which suggests that the highly ordered mesoporous structures were stably preserved even after the Sn addition on the pristine m-CoAlOx.

**Table 1.** Bulk and surface properties of the Sn/m-CoAlOx catalysts such as metal composition, surface area, particle size and reducibility.

| Notation [a] | XRF [b] | N$_2$-Sorption [c] | | | XRD [d] | XPS [e] | H$_2$-TPR [f] | | | CO-Chem. [g] | O$_2$ Titration [g] |
|---|---|---|---|---|---|---|---|---|---|---|---|
| | SnO$_2$ (wt%) | S$_g$ (m$^2$/g) | P$_D$ (nm) | P$_V$ (cm$^3$/g) | Crystallite Size of Co$_3$O$_4$/Co (nm) | ICo/ICo$^{n+}$ (Fresh/Used) | Uptake (mmol/g) | | R$_D$ (%) | S$_{Co}$ (m$^2$/g$_{Co}$) | R$_{DO2}$ (%) |
| | | | | | | | LT | HT | | | |
| m-CoAlOx | 0 | 109.7 | 3.8 | 0.14 | 17.2/12.9 | 0.99/2.51 | 2.64 | 8.67 | 70.5 | 6.6 | 76.1 |
| Sn(2)/m-CoAlOx | 0.25 | 102.7 | 4.0 | 0.14 | 15.7/11.8 | 0.92/2.54 | 2.24 | 7.46 | 66.3 | 6.3 | 72.3 |
| Sn(7)/m-CoAlOx | 0.65 | 105.4 | 4.0 | 0.14 | 14.7/11.0 | 0.92/2.78 | 2.42 | 6.83 | 62.4 | 5.5 | 64.2 |
| Sn(18)/m-CoAlOx | 1.82 | 92.2 | 4.1 | 0.11 | 14.3/10.7 | 0.98/3.03 | 1.52 | 3.82 | 42.2 | 4.0 | 50.2 |

[a] Sn(x)/m-CoAlOx represent the mixed metal oxides of Co$_3$O$_4$-Al$_2$O$_3$ (Co/Al molar ratio of 25) and different amount of Sn promoter with x for wt% of SnO$_2$ × 10. [b] SnO$_2$ content on the Sn/m-CoAlOx was measured by XRF on total weight of catalyst. [c] Specific surface area (S$_g$, m$^2$/g), average pore diameter (P$_D$, nm) and pore volume (P$_V$, cm$^3$/g) of the fresh Sn/m-CoAlOx were measured by N$_2$ adsorption–desorption analysis. [d] Crystallite sizes of the Co$_3$O$_4$ phase on the fresh Sn/m-CoAlOx were calculated by using the most intense diffraction peak at 2θ = 36.8°, and crystallite size of metallic cobalt phase was calculated by using the volume contraction relationship of d(Co$^0$) = 0.75d(Co$_3$O$_4$). [e] Binding energy (BE) of Co 2p$_{3/2}$ peak on the fresh (reduced) and used Sn/m-CoAlOx was measured by XPS analysis, and ICo$^{n+}$/ICo ratio was calculated by using the integrated area of the shoulder peak at ~785 eV (ICo$^{n+}$) and main peak at 780–782 eV (ICo). [f] H$_2$ consumption (uptake) below and above 550 °C was separately denoted as LT and HT, and the degree of reduction (R$_D$, %) was calculated by using the ratio of the experimental H$_2$ uptakes below 550 °C with total hydrogen consumptions for all range of TPR experiments. [g] S$_{Co}$ stands for the metallic surface area of cobalt (m$^2$/g$_{Co}$) on the fresh Sn/m-CoAlOx, and degree of reduction (R$_{DO2}$, %) was calculated by using O$_2$ titration results with the equation of (consumed amount of H$_2$ calculated by O$_2$ titration)/(theoretical amount of H$_2$ consumption) × 100.

The contents of SnO$_2$ on the fresh Sn/m-CoAlOx were found to be close to their nominal values in the range of 0–1.82 wt% as confirmed by XRF results in Table 1. The highly ordered mesoporous structures of the Sn/m-CoAlOx were confirmed by a small angle X-ray scattering (SAXS) analysis as shown in Figure 1A, where the characteristic diffraction peaks appeared at ~0.9° suggest the well-defined ordered mesoporous structures on all the fresh Sn/m-CoAlOx. Those highly ordered mesoporous structures on the Sn/m-CoAlOx were also confirmed by N$_2$ adsorption–desorption analysis, which showed a typical type IV Isotherm with H$_2$ hysteresis as displayed in supplementary Figure S1A. As summarized in Table 1, specific surface area on the Sn/m-CoAlOx was gradually decreased with an increase of Sn content from 109.7 m$^2$/g on the unmodified m-CoAlOx to 92.2 m$^2$/g on the Sn(18)/m-CoAlOx, which were attributed to their selective blockages of mesopore mouths or inner mesoporous surface depositions by adding excess amount of Sn promoter. The average pore diameters on the fresh Sn/m-CoAlOx were found to be in the range of 3.8–4.1 nm (calculated from desorption branch of N$_2$-sorption isotherm) with its small increases due to slight structural collapses at higher Sn contents, which were found to be a little smaller diameter than that of ~4.4 nm calculated from the absorption branch due to the tensile strength and pore network effect [32]. In addition, the pore volume was found to be also similar on all the fresh Sn/m-CoAlOx with 0.14 cm$^3$/g compared to the Sn(18)/m-CoAlOx with the smallest pore volume of 0.11 cm$^3$/g. As displayed in Figure 1B and Supplementary Materials Figure S1B, all the Sn/m-CoAlOx catalysts showed unimodal pore size distributions obtained from N$_2$-adsorption and desorption isotherm, which suggests that the highly ordered mesoporous structures were stably preserved even after the Sn addition on the pristine m-CoAlOx.

Based on the XRD patterns as displayed in Figure 1C on the fresh Sn/m-CoAlOx, the peaks assigned to Co$_3$O$_4$ phases were only observed with its characteristics diffraction peak patterns (JCPDS 42-1467), which suggests that the SnO$_2$ promoter having an intrinsically amorphous nature was well dispersed on the Co$_3$O$_4$-Al$_2$O$_3$ surfaces. The crystallite sizes of the Co$_3$O$_4$ on the fresh Sn/m-CoAlOx frameworks, calculated from the most intense peak

at $2\theta = 36.9°$, were decreased from 17.2 to 14.3 nm with an increase of Sn content, which corresponds to the metallic cobalt crystallite sizes of 12.9–10.7 nm. The observation also suggests that the intrinsic FTS reaction activity can be linearly correlated with the exposed metallic surface area since FTS activity can be assumed as structure insensitive reaction due to the presence of larger particles above 8 nm in size [1,4]. Therefore, the smaller crystallite sizes of cobalt nanoparticles with larger surface areas seem to be beneficial to get a higher catalytic activity, however the exposed surface metal distributions and its electronic states have been reported to be more crucial factors to explain the catalytic activity and stability [1]. In general, Sn promoter can be distributed in the main metal (oxides) structures through the well-known geometric effects of bimetallic catalysts. For example, the Sn atoms can be incorporated in the specific platinum facets by forming ensemble sites, which are beneficial for decreasing coke deposits during $C_2H_6$ dehydrogenation reaction due to the decreased platinum assemble sites [33,34]. As shown in Figure 1D, the ordered mesoporous structures on two typical selected catalysts such as the m-CoAlOx and Sn(7)/m-CoAlOx were clearly observed by TEM images (more magnified images in the Supplementary Materials Figure S2) with the average pore diameter of ~5 nm in size without showing any aggregated or disintegrated mesoporous structures. We believe that the smaller nanoparticles of $Co_3O_4$ phases in the $Co_3O_4$-$Al_2O_3$ matrices with a larger number of metallic surface area as well as their higher reducibility and adsorption natures of syngas are important descriptors for the present structure insensitive FTS reaction with the critical particle size above ~8 nm.

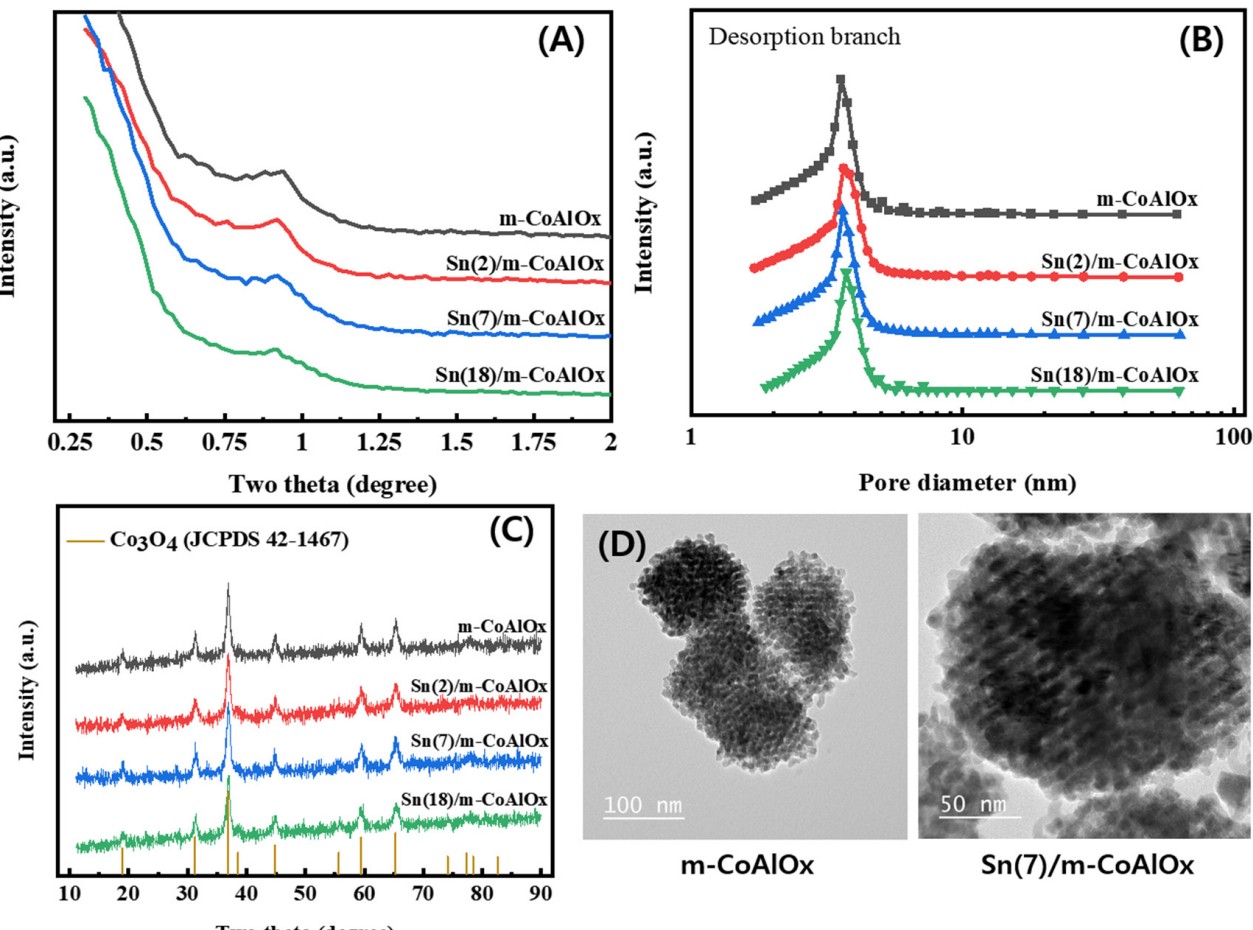

**Figure 1.** Characteristics of the fresh Sn/m-CoAlOx catalysts; (**A**) Small angle XRD patterns, (**B**) Pore size distribution measured by $N_2$ sorption, (**C**) Wide angle XRD patterns and (**D**) TEM images of the m-CoAlOx and Sn(7)/m-CoAlOx.

### 2.2. Reduction Behaviors and Surface Properties of Sn/m-CoAlOx

The reduction behaviors on the fresh Sn/m-CoAlOx catalysts are displayed in Figure 2A, and the $H_2$ consumptions measured from the $H_2$-TPR patterns are summarized in Table 1. The typical reduction peaks of $Co_3O_4$ phases were clearly observed on all the Sn/m-CoAlOx, where first reduction peak at ~345 °C and second broad reduction peak above 480 °C can be attributed to its stepwise reduction through $Co_3O_4 \rightarrow CoO \rightarrow Co^0$ with multiple reduction steps at much higher temperatures due to the different interactions of $Co_3O_4$-$Al_2O_3$ and Sn promoter on the m-CoAlOx [17–22]. Interestingly, the new third reduction peaks were steadily shifted to lower temperatures in the range of 571–629 °C with their larger peak intensity as an increase of Sn content on the Sn/m-CoAlOx. Those observations can be attributed to the possible reduction of CoO phases overcoated with $SnO_2$ promoter or surface reduction stage of $Sn^{4+}$ to $Sn^0$ [35,36] by easily facilitating the reduction of CoO phases to metallic Co nanoparticles. With an increase of Sn content, the first reduction peak (LT) was slightly shifted to a higher temperature from 345 to 362 °C, and the second reduction peak (HT) was inversely shifted to lower temperature from 498 to 444 °C. The observations can be originated from the stronger interactions of $Co_3O_4$ and $SnO_2$ due to the decorations of the active metallic Co surfaces by $SnO_2$ promoter, which are further to suppress $H_2$ adsorption capacity on the Co surfaces and to decrease reduction temperature of the second reduction peak through a possible electron transfer between metallic Co nanoparticles and $SnO_2$ promoter. Therefore, $H_2$ consumption of the first reduction peak (LT) was decreased with an increase of Sn content in the range of 2.64–2.24 mmol/g on the m-CoAlOx and Sn(2)/m-CoAlOx to 1.52 mmol/g on the Sn(18)/m-CoAlOx, where an excess amount of Sn promoter on the Sn(18)/m-CoAlOx caused more significant blockages of the active metallic Co sites. However, $H_2$ consumption of the second reduction peak (HT) was significantly decreased with an increase of Sn content from 8.67 mmol/g on the m-CoAlOx to 3.82 mmol/g on the Sn(18)/m-CoAlOx owing to the relatively even size distributions of $SnO_2$ promoter on the metallic Co surfaces. The surface blockages by $SnO_2$ promoter also decreased the reduction degree ($R_D$), defined as the ratio of $H_2$ consumed amount below 550 °C to total $H_2$ consumption, from 70.5% on the m-CoAlOx to 42.2% on the Sn(18)/m-CoAlOx. These observations can be attributed to the suppression of catalyst reducibility on the Sn/m-CoAlOx, which were also supported by CO-chemisorption analysis with $O_2$ titration as summarized in Table 1. With the decrease of reduction degree on the Sn/m-CoAlOx, the surface areas of metallic cobalt were also decreased from 6.6 to 4.0 $m^2/g_{Co}$ and the degree of reduction ($R_{DO2}$) measured by $O_2$ titration was also decreased from 76.1 to 50.2%, which are in line with the reduction degree ($R_D$) and surface area of metallic Co nanoparticles.

CO adsorption properties on the reduced Sn/m-CoAlOx catalysts were measured by CO-TPD analysis and the results are displayed in Figure 2B. The broad desorption peaks above 350 °C can be attributed to the desorption of CO on the different cobalt phases with their different adsorption strengths as well as the formation of $CO_2$ by a possible Boudouard reaction ($2CO = CO_2 + C$), which were also confirmed by mass spectrometer patterns with $m/z$ = 28 and 44 as displayed in supplementary Figure S3. The disproportionation reaction of CO to form surface carbons and $CO_2$ seems to be kinetically feasible even at a lower temperature above 230 °C [22,23]. With an increase of Sn content on the Sn/m-CoAlOx, the main peak intensity appeared at ~450 °C was significantly decreased due to the possible blockages of the active metallic cobalt sites by the $SnO_2$ promoter resulted in showing a lower catalytic activity compared to the pristine m-CoAlOx, which are in line with the results of TPR and CO-chemisorption.

The surface oxidation states of cobalt nanoparticles with their ratios before and after FTS reaction were measured by XPS analysis, and the typical Co 2p peaks are displayed in Figure 2C and summarized in Table 1. On all the fresh (reduced) Sn/m-CoAlOx, the asymmetrical Co $2p_{2/3}$ peaks were observed at the BEs of 780.1 eV (metallic Co phase) and 790.2 eV (satellite peak), which suggests that the mixed phases of metallic Co and $Co_3O_4$ [37] were mainly formed on the fresh Sn/m-CoAlOx. Interestingly, the Co $2p_{2/3}$

peaks were shifted to the higher BEs of ~781.4 eV on the used Sn/m-CoAlOx, which also suggests that the possible reoxidation of the surface Co species during FTS reaction (or due to air exposure during the characterization) by forming much stronger interactions between $SnO_2$ promoter and $Co_3O_4$ nanoparticles in the Sn/m-CoAlOx frameworks. As displayed in supplementary Figure S4, the Sn species was well distributed on the mesopore channel surfaces of m-CoAlOx below Sn content of 0.65 wt% as supported by its insignificant peak intensity, however, the aggregated $SnO_2$ species on the outer mesopore surfaces was observed on the Sn(18)/m-CoAlOx. In addition, the ratio of oxidation states of Co species on the fresh and used Sn/m-CoAlOx are summarized on Table 1, where the main two deconvoluted peaks appeared at the BEs of ~780 eV and ~784 eV were separately assigned to metallic Co and $Co^{n+}$ species [38,39]. The surface $Co/Co^{n+}$ ratio on the fresh Sn/m-CoAlOx was not significantly altered in the range of 0.92–0.99 due to a hard reducibility of $Co_3O_4$ under the present reduction condition. However, the surface $Co/Co^{n+}$ ratio on the used Sn/m-CoAlOx was largely changed in the range of 2.51–3.03 due to the reduction of the outermost surface Co species by forming the dominant bulk $Co_3O_4$ phases. In addition, the phenomena were found to be more significant on the excess Sn-modified Sn(18)/m-CoAlOx with the larger ratio of 3.03 due to the much stronger interactions between the aggregated $SnO_2$ promoter and $Co_3O_4$ nanoparticles by showing the much smaller meal surface area $(4.0 \text{ m}^2/\text{g}_{Co})$ of metallic Co sites.

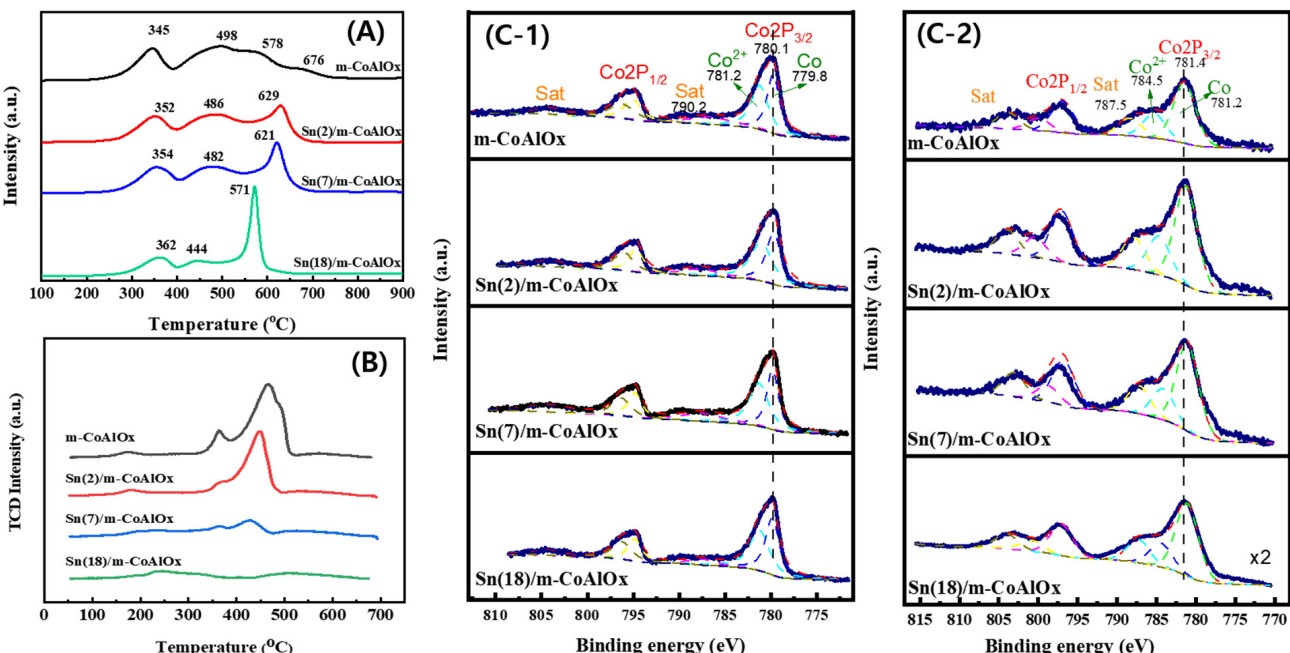

**Figure 2.** Bulk and surface propeties of the fresh Sn/m-CoAlOx catalysts; (**A**) TPR patterns, (**B**) CO-TPD profiels, and (**C**) XPS spectra of Co $2p_{3/2}$ peak on the (**C-1**) fresh (reduced) and (**C-2**) used Sn/m-CoAlOx.

## 2.3. Catalytic Activity and Product Distribution on the Sn/m-CoAlOx Catalysts

The catalytic activity and hydrocarbon distribution on the Sn/m-CoAlOx were measured under the typical FTS reaction conditions of T = 230 °C, P = 2.0 MPa, and space velocity (SV) = 8000 L/(kg$_{cat}$·h) for 60 h on stream as shown in Figure 3A, and steady-state catalytic activity and product distribution are summarized in Table 2. With an increase of Sn content on the Sn/m-CoAlOx, the steady-state CO conversion was slightly decreased from 81.5% on the m-CoAlOx to 71.5% on the Sn(7)/m-CoAlOx and its dramatic decrease down to 52.7% on the Sn(18)/m-CoAlOx. Those activity variations were mainly attributed to the excess blockages of the active metallic cobalt sites by $SnO_2$ promoter as confirmed by TPR, CO-chemisorption and XPS analysis. Interestingly, the induction period to ap-

proach steady-state CO conversion was suppressed with an increase of Sn content, which suggests that the $SnO_2$ promoter seems to selectively block the active Co sites and to preferentially adjust the surface reconstruction of the active metallic Co surfaces [1,6]. The different amount of the exposed surface metallic Co sites according to Sn content on the Sn/m-CoAlOx surfaces largely altered the adsorption capacities of CO and $H_2$ molecules due to the different reducibility of $Co_3O_4$ nanoparticles, which further changed the catalytic activity and hydrocarbon distribution. The larger content of the metallic Co sites originated from a larger reducibility on the pristine m-CoAlOx without $SnO_2$ promoter was responsible for an enhanced adsorption capacity of CO and $H_2$ reactant, which were resulted in an enhanced CO conversion of 81.5% as well as a higher $CH_4$ and $C_2$–$C_4$ selectivity of 24.0 and 29.2% with lower $C_{5+}$ selectivity of 46.8% ($C_{5+}$ yield of 35.3% and $\alpha$ = 0.786). With an increase of Sn content on the Sn/m-CoAlOx, CO conversions were steadily decreased to 79.1–52.7% with the suppressed $CH_4$ and $C_2$–$C_4$ selectivity of 23.3–16.4 and 27.7–20.8% and an increased $C_{5+}$ selectivity of 49.0–62.8% ($\alpha$ = ~0.79). The yields of $C_{5+}$ hydrocarbons were maximized on the Sn(2)/m-CoAlOx and Sn(7)/m-CoAlOx in the range of 36.0–38.1%, which seems to be attributed to the selective blockages of the much active Co sites by $SnO_2$ promoter resulted in the decreased CO conversion to $CO_2$ from 7.4% on the m-CoAlOx to 1.5% on the Sn(18)/m-CoAlOx due to the suppressed WGS reaction activity.

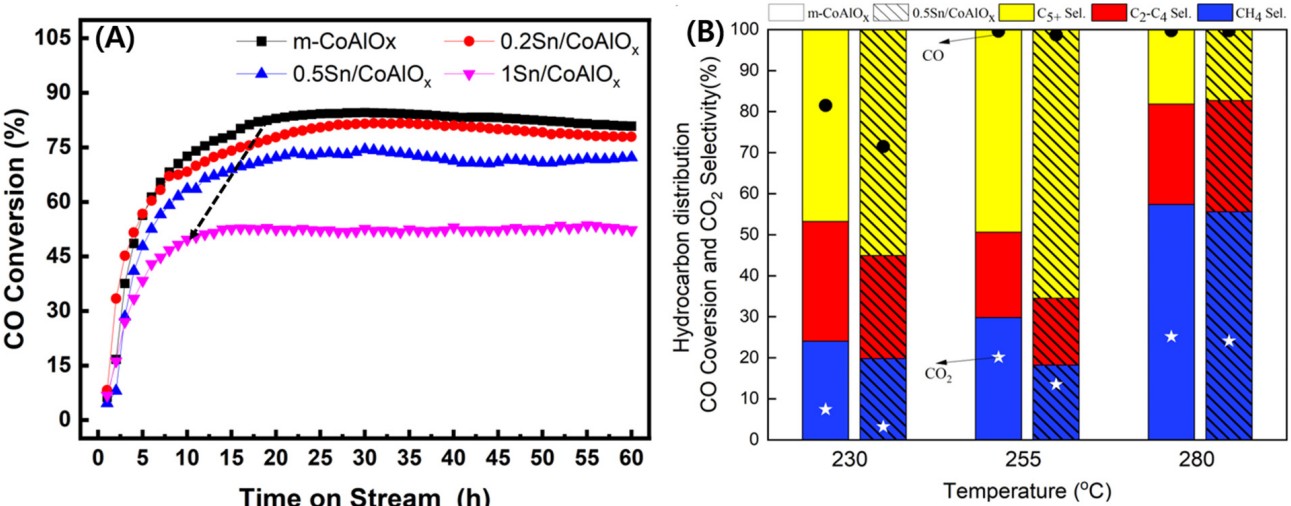

**Figure 3.** (**A**) CO conversion on the Sn/m-CoAlOx catalysts and (**B**) Temperature effects to product distribution on the pristine m-CoAlOx (undashed histogram) and Sn(7)/m-CoAlOx (dashed histogram).

To further verify the roles of the Sn promoter for the product distributions, catalytic activity on two selected catalysts such as the pristine m-CoAlOx and Sn-modified Sn(7)/m-CoAlOx was compared at three different reaction temperatures of 230–255–280 °C, and the results are displayed in Figure 3B as well as in Supplementary Materials Figure S5 and Table S1. At the increased temperatures above 255 °C, CO conversions approached to 99.6% (81.5% @230 °C) and 98.7% (71.5% @230 °C) on the m-CoAlOx and Sn(7)/m-CoAlOx, respectively. Interestingly, the suppressed $CO_2$ selectivity with 6.6% formed by WGS reaction and $CH_4$ selectivity with 11.6% ($C_{5+}$ selectivity above 65.5%) were observed on the Sn(2)/m-CoAlOx compared to those on the pristine m-CoAlOx (respective $CO_2$ and $CH_4$ selectivity of 20.1 and 29.8%) at the reaction temperature of 255 °C. At the much higher temperature of 280 °C, the differences of the $CO_2$ and $CH_4$ selectivity were found to be much smaller between the m-CoAlOx and Sn(7)/m-CoAlOx with their respective differences of 1.0 and 1.8%. The observations suggest that the roles of $SnO_2$ promoter on the Sn/CoAlOx are to effectively suppress the undesired $CO_2$ and $CH_4$ formations at an appropriate temperature of ~250 °C and the proper amount of Sn promoter in the range of 0.25–0.65 wt% by slightly decreasing CO conversion. Those positive contributions of $SnO_2$

promoter were attributed to their selective depositions on the highly active cobalt sites, which were responsible for the less formations of underside $CH_4$ and $CO_2$ by WGS reaction.

**Table 2.** Catalytic activity and product distribution on the Sn/m-CoAlOx catalysts [a].

| Catalyst | CO Conv. (C-mol%) | CO Conv. to $CO_2$ (C-mol%) | Product Distribution (C-mol%) | | | Olefins in $C_2$–$C_4$ (%) | $C_{5+}$ Yield (%) | $\alpha$ [b] |
|---|---|---|---|---|---|---|---|---|
| | | | $C_1$ | $C_2$–$C_4$ | $C_{5+}$ | | | |
| m-CoAlOx | 81.5 | 7.4 | 24.0 | 29.2 | 46.8 | 37.1 | 35.3 | 0.786 |
| Sn(2)/m-CoAlOx | 79.1 | 7.0 | 23.3 | 27.7 | 49.0 | 32.3 | 36.0 | 0.797 |
| Sn(7)/m-CoAlOx | 71.5 | 3.2 | 19.8 | 25.1 | 55.1 | 24.0 | 38.1 | 0.795 |
| Sn(18)/m-CoAlOx | 52.7 | 1.5 | 16.4 | 20.8 | 62.8 | 37.6 | 32.6 | 0.799 |

[a] Catalytic performances were measured at the reaction conditions of T = 230 °C, P = 2.0 MPa, space velocity (SV) = 8000 L/(kg$_{cat}$·h) and feed gas of $H_2$/$N_2$/CO = 63.0/31.5/5.5 (mol%). [b] Chain growth probability of hydrocarbons, calculated from Anderson-Schulz-Flory (ASF) distribution above $C_{5+}$ hydrocarbons formed, was represented by $\alpha$ value.

## 3. Experimental Section

### 3.1. Catalyst Preparation and Activity Measurement

The ordered mesoporous $Co_3O_4$-$Al_2O_3$ bimetal oxides were prepared by using a hard template of KIT-6 based on previous works [17,18,21] and the Sn-modified $Co_3O_4$-$Al_2O_3$ bimetal oxides were prepared by an incipient-wetness impregnation method with different Sn content. The ordered mesoporous KIT-6 template was previously synthesized by using an amphiphilic triblock copolymer (poly(ethyleneoxide)-poly(propyleneoxide)-poly(ethyleneoxide), Pluronic P123, $EO_{20}PO_{70}EO_{20}$, Sigma Aldrich) as a structure-directing agent and tetraethyl orthosilicate (TEOS, Alfa Aesar) as a silica source at 110 °C for 24 h. For more details, 20 g of P123, 20 g of n-butanol and 30 mL HCl solution were dissolved in a deionized water (DIW) with 500 mL under vigorous stirring conditions at 35 °C for 1 h, and 40 g of TEOS was successively added to the above solution for 24 h. After a vigorous stirring, the gel was transferred to an autoclave for a hydrothermal synthesis at 110 °C for 24 h at a static condition, which was washed and filtered with DIW several time and dried at 100 °C overnight and calcined at 550 °C for 6 h at a heating rate of 1 °C/min under air environment to prepare KIT-6 hard template. Successively, 15.05 g of $Co(NO_3)_2$·$6H_2O$ and 4.871 g of $Al(NO_3)_3$·$9H_2O$ precursor were dissolved in DIW and the solution was impregnated into 10 g of KIT-6 by an incipient-wetness impregnation method. The metal precursors-incorporated KIT-6 hard template was dried at 80 °C overnight, and calcined at 550 °C for 3 h at a heating rate of 1 °C/min. In order to remove the KIT-6, the as-prepared mixed metal oxides were refluxed in 2M NaOH solution at 70 °C for 3 h, and washed with DIW several time followed by drying at 80 °C overnight, where the final catalyst was denoted as the m-CoAlOx. The Sn-modified m-CoAlOx with different Sn content in the range of 0.25–1.82 wt%Sn was prepared by an incipient wetness impregnation method. For more details, the desired nominal amount of $SnCl_2$·$2H_2O$ precursor diluted in ethanol solvent was impregnated into the as-prepared m-CoAlOx, and it was dried at 80 °C and calcined at 400 °C for 4 h again. The Sn-modified ordered mesoporous m-CoAlOx were denoted as Sn(x)/m-CoAlOx, where x denotes Sn content (wt%) based on the total weight of pristine m-CoAlOx.

The catalytic activity and product distribution on the Sn/m-CoAlOx were measured by using a fixed-bed tubular stainless-steel reactor having an outer diameter of 12.7 mm. For more details, 0.1 g of catalyst with 1 g of inert $\alpha$-$Al_2O_3$ as an inert material was physically mixed using a mortar agate and loaded into the reactor. Prior to FTS reaction, the catalyst was reduced under a flow of 30 mL/min of 5%$H_2$/$N_2$ at 400 °C for 12 h at a heating rate of 5 °C/min. Subsequently, feed gas composed of $H_2$/CO/$N_2$ = 63.0/31.5/5.5 (molar ratio) was introduced into the reactor at the fixed reaction conditions of T = 230 °C, P = 2.0 MPa, weight hourly space velocity (SV) of 8000 L/(Kg$_{cat}$·h) for 60 h on stream. The effluent gases were analyzed by using an on-line gas chromatography (GC, YoungLin GC 6000) to analyze gas-phase products and by using an off-line GC to analyze liquids and wax products, which were collected in a cold trap maintained at 60 °C. For an on-line GC

operation, a thermal conductivity detector (TCD) with Carboxen 1000 packed column was used for analyzing $N_2$, CO, $CO_2$, $CH_4$ and $H_2$ gas and a flame ionization detector (FID) with GS-GASPRO capillary column was used for analyzing $C_1$–$C_9$ hydrocarbons. For an off-line GC operation, FID with HP-5 capillary column was used for analyzing liquid hydrocarbons and the hydrocarbon distributions and CO conversion on the Sn/m-CoAlOx were calculated based on total carbon balances.

### 3.2. Characterization

The crystalline phases on the fresh Sn/m-CoAlOx were obtained by using X-ray diffraction (XRD) patterns with an X'Pert PRO MPD (PANalytical) instrument equipped with a Cu-K$\alpha$ radiation ($\lambda$ = 0.15418 nm) operated at 40 kV and 30 mA in the range of $2\theta$ = 10–90°. The crystallite sizes of $Co_3O_4$ phases on the fresh Sn/m-CoAlOx were also calculated by using the most intense diffraction peak at $2\theta$ = 36.8° with the help of Scherrer equation, and the crystallite sizes of metallic cobalt phases were calculated by using the volume contraction relationship of $d(Co^0) = 0.75d(Co_3O_4)$. X-ray fluorescence (XRF) analysis was carried out to elucidate the bulk compositions of the fresh Sn/m-CoAlOx by using an Axios Minerals (PANalytical) instrument operated at 60 kV and 125 mA. The ordered mesoporous structures on the fresh Sn/m-CoAlOx were also confirmed with a small angle X-ray scattering (SAXS) analysis and those mesoporous morphologies were further confirmed by a high-resolution transmission electron microscopy (HRTEM) with a JEM-2100F(JEOL) equipment.

$N_2$ adsorption–desorption analysis was carried out to confirm the textural properties of the fresh Sn/m-CoAlOx by using a Tristar II (Micromeritics) instrument at liquid $N_2$ temperature of −196 °C. The specific surface area was calculated by using Brunauer-Emmett-Teller (BET) method, and pore size distribution was obtained by Barrett-Joyner-Halenda (BJH) model from the adsorption and desorption isotherm, respectively. Prior to the analysis, all fresh samples were degassed at 350 °C for 4 h under a vacuum condition to remove any impurity and water adsorbed. The surface area of metallic cobalt nanoparticles ($m^2/g_{Co}$) on the fresh Sn/m-CoAlOx was measured by CO chemisorption by using a Micromeritics ASAP 2000 equipment at a static condition of 100 °C. For more details, 0.1 g of sample was degassed at 350 °C for 4 h under a vacuum condition, and it was reduced under a 10% $H_2$/Ar flow (30 mL/min) at 400 °C for 12 h followed by CO chemisorption at 100 °C. In addition, successive $O_2$ titration was carried out after CO chemisorption, and the metallic cobalt surface area was calculated with an assumption of CO/Co stoichiometry of 1.0 and the degree of reduction ($R_{DO2}$, %) was calculated by using $O_2$ titration result based on the theoretically required total amount of $O_2$ consumption.

Temperature-programmed reduction with hydrogen (TPR) was carried out to confirm the reduction behaviors of the fresh Sn/m-CoAlOx by using a BELCAT equipment with a TCD detector. For more details, 7 mg sample was loaded in a cell and pretreated under Ar flow at 30 mL/min at 400 °C for 1 h to remove contaminants and water adsorbed on the catalyst surfaces. After colling down it to 100 °C, reduction patterns were obtained up to 900 °C at a heating rate of 10 °C/min under a flow rate of 30 mL/min of 10% $H_2$/Ar gas. Degree of reduction ($R_D$, %) of the fresh Sn/m-CoAlOx was also calculated by using the ratios of experimental $H_2$ uptakes below 550 °C to total $H_2$ consumption for the full TPR temperature range.

The adsorption properties of CO molecules on the active cobalt sites of the Sn/m-CoAlOx were measured by using temperature-programmed desorption of CO (CO-TPD) with a BELCAT equipment combined with a quadrupole mass spectrometer (MS) as well. For the CO-TPD analysis, 0.1g sample was loaded in the cell and reduced at 400 °C with 10% $H_2$/Ar for 1 h, which was purged out with a carrier He gas at 230 °C for 30 min to remove adsorbed $H_2$ on the Sn/m-CoAlOx surfaces. The reduced catalyst was exposed to pure CO gas (50 mL/min) at 230 °C for 1 h and cooled down to 50 °C, and flushed with He for 1 h to remove all physically adsorbed CO molecules. TPD analysis was performed by heating up to 900 °C at a ramping rate of 10 °C under He flow, which were analyzed with

TCD and MS simultaneously. X-ray photoelectron spectroscopy (XPS) was also performed by using K-Alpha XPS system (Thermo Fisher Scientific) to analyze the outermost surface chemical species as well as oxidate states of the fresh (reduced) and used Sn/m-CoAlOx. The binding energy (BE) was corrected with several reference oxidation states of cobalt socies by using Avantage program and further corrected by using the reference BE of C1s at 284.6 eV. Before the XPS analysis, all the used Sn/m-CoAlOx were washed several times by using hexane solvent to remove waxy hydrocarbons deposited on the Sn/m-CoAlOx surfaces during FTS reaction.

## 4. Conclusions

To mitigate the product distribution with smaller $CH_4$ and $CO_2$ formation during FTS reaction, the highly ordered mesoporous $Co_3O_4$-$Al_2O_3$ bimetal oxide modified with a proper amount of Sn promoter in the range of 0.25–0.65 wt% (Sn/m-CoAlOx) revealed an enhanced selectivity of liquid-hydrocarbons with small decrease of CO conversion due to non-selective blockages of active cobalt sites by Sn promoter. The more difficult reducibility of cobalt nanoparticles and suppressed number of active metallic cobalt sites were observed with an increase of Sn content above 1.82 wt%, which were responsible for the decreased CO conversion as well. The decreased $CO_2$ and $CH_4$ selectivity was clearly observed on the Sn(2)/m-CoAlOx with small decrease of CO conversion with 79.1% from 81.5% on the reference m-CoAlOx, which were mainly attributed to the suppressed WGS reaction activity and decreased full hydrogenation activity to form $CH_4$ due to the suppressed reducibility of the partially reduced $Co_3O_4$ phases with the help of $SnO_2$ promoter.

**Supplementary Materials:** The following supporting information can be downloaded at: https://www.mdpi.com/article/10.3390/catal12111447/s1, Figure S1 for characteristics of the fresh Sn/m-CoAlOx catalysts; (A) $N_2$ adsorption-desorption isotherms and (B) Pore size distribution obtained from adsorption branch; Figure S2 for TEM images of the (A) unmodified m-CoAlOx and (B) Sn(7)/m-CoAlOx; Figure S3 for CO-TPD profiles on the reference m-CoAlOx catalyst (Figure 2B), which were obtained from TCD and mass spectrometer with $m/z$ = 28 (CO) and 44 ($CO_2$); Figure S4 for XPS spectra of Sn 3d5/2 peak on the fresh (reduced) Sn/m-CoAlOx; Figure S5 for temperature effects to product distributions (Figure 3B) on the (A) m-CoAlOx and (B) Sn(7)/m-CoAlOx; Table S1 for temperature effects to product distributions (Figure 3B) on the (A) m-CoAlOx and (B) Sn(7)/m-CoAlOx.

**Author Contributions:** Methodology, D.S.; validation, S.B.H. and D.S.; formal analysis, D.S.; data curation, X.W. and M.A.; writing—original draft preparation, D.S. and S.B.H.; writing—review and editing, J.W.B.; supervision, J.W.B.; funding acquisition, J.W.B. All authors have read and agreed to the published version of the manuscript.

**Funding:** Korea Electric Power Corporation of the Republic of Korea (Grant number: R21XA01-29) and National Research Foundation of Korea (NRF, 2021R1A4A1024129).

**Conflicts of Interest:** The authors declare no conflict of interest.

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
