# Peer review of "Effects of Sn Promoter on the Ordered Mesoporous Co3O4-Al2O3 Mixed Metal Oxide for Fischer–Tropsch Synthesis Reaction"

_catalysts, doi:10.3390/catal12111447_

Round 1

Reviewer 1 Report

My comments are in the PDF document.

Author Response

Reviewer(#1 and 2)'s comments are attaced. 

Reviewer 2 Report

In this manuscript, authors reported the synthesis of porous Co3O4-Al2O3 metal oxides modified with different amounts of Sn and studied their catalytic performance for Fischer–Tropsch Synthesis. Overall, the manuscript is well-written, and the conclusions are supported by the data. I recommend minor revision before publication.
Nanocasting is a common procedure to prepare materials with ordered pore structures, it will be better if the motivation for using nanocasting can be discussed in the introduction. Also, is there a specific reason why KIT-6 is used as the template? Have the authors considered other materials such as SBA-15 or SBA-16?
Was the offset applied for the isotherms in Figure S1A? If so, the offset should be provided in the caption, and the y-axis needs to be added.
What is the distribution of Sn species in the bimetal oxide? Is the Sn on the outside of the bimetal oxide particles or everywhere inside the mesoporous channels? And why does the pore size increase from 3.8 nm to 4.0 or 4.1 nm after Sn modification?

Author Response

(The authors gave the same response as above.)

Round 2

Reviewer 1 Report

The article has been improved